



# Comprehensive Quantification of Height Dependence of Entrainment-Mixing between Stratiform Cloud Top and Environment

Sinan Gao[1], Chunsong Lu[1]*, Yangang Liu[2], Seong Soo Yum[3], Jiashan Zhu[1], Lei Zhu[1], Neel Desai[2a], Yongfeng Ma[4]

[1]Collaborative Innovation Center on Forecast and Evaluation of Meteorological Disasters, Key Laboratory for Aerosol-Cloud-Precipitation of China Meteorological Administration, Nanjing University of Information Science & Technology, Nanjing, China

[2]Environmental and Climate Sciences Department, Brookhaven National Laboratory, Upton NY, US

[3]Department of Atmospheric Sciences, Yonsei University, Seoul, South Korea

[4]Department of Mechanics & Aerospace Engineering, Southern University of Science and Technology, Shenzhen, China

*Correspondence to*: Chunsong Lu (luchunsong110@gmail.com)

---

[a] Now at Department of Meteorology and Climate Science, San Jose State University, San Jose, CA



**Abstract.** Different entrainment-mixing processes of turbulence are crucial to processes related to clouds; however, only a few
qualitative studies have been concentrated on the vertical distributions of entrainment-mixing mechanisms with low vertical
resolutions. To quantitatively study vertical profiles of entrainment-mixing mechanisms with a high resolution, the stratiform
clouds observed in the Physics of Stratocumulus Top (POST) project are examined. The unique sawtooth flight pattern allows
for an examination of the vertical distributions of entrainment-mixing mechanisms with a 5 m vertical resolution. Relative
standard deviation of volume mean radius divided by relative standard deviation of liquid water content is introduced to be a
new estimation of microphysical homogeneous mixing degree, to overcome difficulties of determining the adiabatic
microphysical properties required in existing measures. The vertical profile of this new measure indicates that entrainment-
mixing mechanisms become more homogeneous with decreasing altitudes and are consistent with the dynamical measures of
Damkohler number and transition scale number. Further analysis shows that the vertical variation of entrainment-mixing
mechanisms with decreasing altitudes is due to the increases of turbulent dissipation rate in cloud and relative humidity in
droplet-free air, and the decrease of size of droplet-free air. The results offer insights into the theoretical understanding and
parameterizations of vertical variation of entrainment-mixing mechanisms.



## 1 Introduction

Clouds are identified to be a significant origin of uncertainties in climate research, because of poor simulations of clouds (Bony and Dufresne, 2005; Stephens, 2005; Zheng and Rosenfeld, 2015; Zhao and Garrett, 2015; Wang et al., 2019; Cess et al., 1989; Wang, 2015; Gao et al., 2016; Grabowski, 2006; Morrison, 2015). Entrainment-mixing processes of turbulence have been considered as significant factors for various processes related to clouds (Su et al., 1998; Lasher‐trapp et al., 2005; Hoffmann and Feingold, 2019; Xu et al., 2020; Hudson et al., 1997; Liu et al., 2002). Therefore, it is vital to figure out the nature of interaction between clouds and environment and their impacts on cloud droplet properties (Xue and Feingold, 2006). Entrainment-mixing processes are considered to occur primarily near the stratiform cloud top and entrainment-mixing around the stratiform cloud sides is negligible (Wood, 2012; Xu and Xue, 2015).

The question about how entrained air affects cloud microphysics has been debated for a long time. Several conceptual models have been established to study the different entrainment-mixing processes, e.g., entity-type entrainment-mixing (Telford, 1996; Telford and Chai, 1980), vertical circulation entrainment-mixing (Yeom et al., 2017; Yum et al., 2015; Wang et al., 2009) and homogeneous (HM)/inhomogeneous (IM) entrainment-mixing (Baker et al., 1980; Baker et al., 1984). The last one is the most used and studied. During the HM mixing, the time scale for droplets to evaporate completely is larger than the time scale for mixing between entrained air and cloudy air. All droplets are exposed to the same unsaturated state and evaporate concurrently. In this scenario, all droplets' sizes decrease simultaneously, and number concentration also decreases due to the dilution effect of entrained air. While in the IM mixing, mixing time scale is larger than evaporation time scale. Some droplets adjacent to entrained air would evaporate completely to saturate the air, while the other droplets are not affected by the entrainment. In this scenario, number concentration decreases but droplet size remains unchanged. Some observational studies support the extreme IM concept (Burnet and Brenguier, 2007; Lu et al., 2011; Freud et al., 2011; Pawlowska et al., 2000; Haman et al., 2007; Freud et al., 2008); while some others indicate that the HM mixing dominates (Gerber et al., 2008; Lu et al., 2013c; Burnet and Brenguier, 2007; Jensen et al., 1985), and still some others find intermediate features fall in between the HM and IM mixing (Lehmann et al., 2009; Lu et al., 2014a; Kumar et al., 2018).

The vertical variation of entrainment-mixing mechanisms is less studied. For cumulus, Small et al. (2013) and Jarecka et al. (2013) found that a trend existed of entrainment-mixing to be more HM in cloud top, resulted from increasing of cloud droplet radius and turbulence with increasing altitudes. In stratiform clouds, Yum et al. (2015) and Wang et al. (2009)observed positive correlation at middle of cloud and no correlation at cloud top between droplet size and liquid water content. Yum et al. (2015) suggested that entrainment mixing at cloud top region was indeed IM, while during the descent of vertical circulation, the cloud droplets in more diluted parcels would evaporate faster, and observe the generally HM feature at a relatively long depth



from cloud top.

The above few studies are largely qualitative and based on horizontal flight legs with coarse vertical resolutions. Furthermore,
these studies often need to determine adiabatic cloud microphysical properties from observational data, which are full of known
and unknown uncertainties (e.g., (Jensen et al., 1985; Yum et al., 2015; Lu et al., 2014b; Yeom et al., 2017).

This study aims to overcome these limitations by examining the data from the field campaign of Physics of Stratocumulus Top
(POST) (Hill et al., 2010; Malinowski et al., 2010; Gerber et al., 2010) for the high-resolution vertical variation of entrainment-
mixing processes. Four measures of microphysical homogeneous mixing degrees (HMDs) that require the determination of
adiabatic cloud properties (Lu et al., 2014b; Lu et al., 2013b; Lu et al., 2014a) are examined and inconsistencies are discussed.
A new microphysical measure is proposed to quantify the entrainment-mixing mechanisms to overcome the drawbacks of the
existing methods that require cloud adiabatic properties. Physical reasons for the vertical variation of entrainment-mixing
mechanisms are analyzed using a comprehensive microphysical-dynamical approach.

The rest of this study is presented as follows. The POST dataset and the existing methods for calculating microphysical and
dynamical measures of HMD are presented in Section 2. Section 3 first shows the analysis of entrainment-mixing mechanisms
using the existing microphysical measures and dynamical measures. A new microphysical measure is then introduced to
represent entrainment-mixing mechanisms after discussing the potential uncertainties in choosing and determining the
adiabatic properties needed for the existing microphysical measures. The key factors affecting vertical variation of
entrainment-mixing are examined as well. Section 4 is the concluding remarks.
**2 Dataset and Methods**
**2.1 Dataset**
POST was designed to further the understanding of the physical processes around stratiform cloud top zone (Carman et al.,
2012; Gerber et al., 2010; Hill et al., 2010; Malinowski et al., 2010; Ma et al., 2017; Jen-La Plante et al., 2016; Ma et al., 2018;
Kumala et al., 2013). During POST campaign, thermodynamic, dynamical, and microphysical properties were measured on
board in July and August of 2008 with a total of 17 research flights. Flights were implemented in the vicinity of the coast of
Santa Cruz/Monterey, California, US, within $36°$ to $37°$N and$123°$ to $124°$W(Gerber et al., 2010; Hill et al., 2010; Malinowski
et al., 2010).

Cloud droplet distributions were from the measurement by the Cloud Aerosol Spectrometer (CAS) probe, and the measured



frequency is 10 Hz. The microphysical properties, number concentration ($n_c$), liquid water content ($LWC_c$) and volume mean
radius ($r_{vc}$) are calculated from the cloud droplet distributions using the radius range of 1 - 25 μm. The Modified Ultrafast
Thermometer (UFT-M) was the temperature probe. Only the flights with good quality temperature data (no reports of "noise",
"spike" or "holes in the data" in the data description file) are used. Although the time resolution of temperature data was as
high as 1000 Hz (Kumala et al., 2013), 10 Hz data are used here. Humidity was measured by the EDGETECH EG&G Chilled
Mirror at 10 Hz. For turbulence measurements, the five-hole gust detector provided by University of California, Irvine (UCI)
was used to collect high resolution wind velocities at 40 Hz. We use 10 cm$^{-3}$ of $n_c$ and 0.001 g m$^{-3}$ of $LWC_c$ to be the standard
of threshold values to select cloudy samples (Lu et al., 2014b; Deng et al., 2009; Zhang et al., 2011). We define the cloud base
as the lowest altitudes where the samples satisfy the previously mentioned cloud criteria. We focus only on the non-drizzling
clouds, and the threshold value of drizzle water content in cloud using Cloud Imaging Probe (CIP) measurements (radius larger
than 25 μm) is 0.005 g m$^{-3}$ (Lu et al., 2011). A total of 4 flights in POST (July 16, August 02, 06, 08, 2008) satisfying the above
criteria is selected to examine the vertical variation of entrainment-mixing mechanisms.
**2.2 Sawtooth Pattern Flights**
Unlike most aircraft campaigns, the POST flights were designed as sawtooth legs to examine detailly the vertical structures of
the stratiform cloud top zone (Figure 1 (a)) (Carman et al., 2012; Gerber et al., 2013; Jen-La Plante et al., 2016). About 60
sawtooth legs are contained in each flight (Gerber et al., 2013; Carman et al., 2012). In this way, high-resolution vertical
profiles near cloud top can be obtained, which are not available from the conventional sampling along horizontal legs. Because
the cloud top altitudes vary spatially, we calculate the average cloud top altitude measured by each sawtooth profile and only
the sawtooth legs with cloud tops 30 m above/below the average cloud top are selected. The procedure of altitude stratification
is illustrated in Figure 1 (b). We take 5 m as the vertical interval of all sawtooth patterns. All the analyses below are based on
the cloud properties averaged over the 5 m vertical intervals and each vertical interval consists of thousands of data. Only the
height intervals over which the average droplet-free air sizes (i.e., non-cloudy sample sizes between cloudy samples) are larger
than zero are analyzed, which is detailed later in Figure 10. The results are similar when the vertical resolution of all sawtooth
patterns is set as 3 m and 7 m, respectively (not shown).
**2.3 Methods**
**2.3.1 Existing Microphysical Measures of Homogeneous Mixing Degree**
Based on the diagram of microphysical mixing, four HMDs have been defined to contain all kinds of entrainment mixing
mechanisms. The first three measures are based on the diagram of $r_{vc}^3/r_{va}^3$ versus $n_c/n_a$ (Lu et al., 2014a; Lu et al., 2013b), as



shown in Figure 2 (a) and (b). Figure 2 (a) declares the various status during a whole process of entrainment-mixing for
defining the first measure ($\psi_1$). The adiabatic cloud is represented by Point A with the number concentration ($n_a$) and volume
mean radius ($r_{va}$) of adiabatic state. After environmental air is entrained into cloud, the state of cloud approaches Point B,
where number concentration is $n_h$ and volume mean radius is $r_{va}$. Then mixing and evaporation processes occur and cloud state
approaches Point C, where number concentration after evaporation is $n_c$ and volume mean radius after evaporation is $r_{vc}$. The
included angle between the line connecting Point B to Point E and the extreme IM mixing line is $\pi/2$, and the included angle
between the line connecting Point B to Point C and the extreme IM mixing line is $\beta$. Then $\psi_1$ is defined as:
$$\psi_1 = \frac{\beta}{\pi/2},$$ (1a)
where $\beta$ is
$$\beta = \arctan(\frac{r_{vc}^3/r_{va}^3 - 1}{n_c/n_a - n_h/n_a});$$ (1b)
$n_h = n_a \times \chi$ and $\chi$ represents the adiabatic cloud fraction after mixing derived from energy conservation and total water
conservation in the isobaric mixing (Lehmann et al., 2009; Gerber et al., 2008; Lu et al., 2012). The second HMD ($\psi_2$) is
defined in view of Figure 1 (b):
$$\psi_2 = \frac{1}{2}(\frac{n_c - n_i}{n_h - n_i} + \frac{r_{vc}^3 - r_{va}^3}{r_{vh}^3 - r_{va}^3}),$$ (2)
where $$n_i = \frac{r_{vc}^3}{r_{va}^3}n_c.$$ (3)
$$r_{vh}^3 = \frac{n_c}{n_h}r_{vc}^3 \quad \text{and}$$ (4)
Here $n_i$ is the number concentration after extreme IM mixing and $r_{vh}$ is the volume mean radius after HM mixing. The third
measure of HMD ($\psi_3$) is given by
$$\psi_3 = \frac{\ln n_c - \ln n_i}{\ln n_h - \ln n_i} = \frac{\ln r_{vc}^3 - \ln r_{va}^3}{\ln r_{vh}^3 - \ln r_{va}^3}.$$ (5)
The fourth measure ($\psi_4$) is defined using mixing diagram of $r_{vc}^3/r_{va}^3$ versus $LWC_c/LWC_a$ (Lu et al., 2014b), as shown in Figure
2 (c),
$$\psi_4 = \frac{1 - r_{vc}^3/r_{va}^3}{1 - LWC_c/(\chi LWC_a)}.$$

139 (6)

The meanings of the Points A - E are the same as those in Figures 2 (a) and 2 (b). Four kinds of HMDs are expected to range
from 0 to 1, the higher probability of HM mixing corresponds to the larger HMD value.





**2.3.2 Dynamical Measures of Homogeneous Mixing Degree**
The dynamical aspect, i.e., the mixing process between cloud and environment air *vs.* the evaporation process of cloud droplets,
is important to distinguish different entrainment-mixing mechanisms (Baker et al., 1980; Baker and Latham, 1979). The mixing
time scale divided by evaporation time scale is defined as Damkohler number ($Da$), which is usually used to quantify mixing
process is faster or evaporation process is faster and thus to discern the entrainment-mixing mechanisms (Siebert et al., 2006;
Burnet and Brenguier, 2007; Andrejczuk et al., 2009),
$$Da = \frac{\tau_{mix}}{\tau_r},$$  (7)
where $\tau_{mix}$ and $\tau_r$ are turbulent mixing time and microphysical response time of droplets, respectively (Lehmann et al., 2009).
A more IM mixing corresponds to a larger $Da$. Three kinds of microphysical time scales, phase relaxation time ($\tau_{phase}$) (Kumar
et al., 2013; Kumar et al., 2012), evaporation time ($\tau_{evap}$) (Andrejczuk et al., 2009; Baker et al., 1980; Burnet and Brenguier,
2007), and reaction time ($\tau_{react}$) (Lehmann et al., 2009; Lu et al., 2011; Lu et al., 2013c; Lu et al., 2014b), have been used to
represent $\tau_r$. Lu et al. (2018) found that the most appropriate time scale was $\tau_{evap}$ if we focus on the changes of number
concentration and radius of droplets. The mixing time scale is defined as follows:
$$\tau_{mix} \sim (L^3/\varepsilon)^{1/3},$$  (8)
where $\varepsilon$ is the turbulent dissipation rate calculated from the three dimensional wind velocities (Meischner et al., 2001) (see
Appendix A for details), and $L$ is the size of droplet-free air calculated with
$$L = F \times TAS/f,$$  (9)
where droplet-free sample size divided by the sum of cloud and droplet-free sample size is considered as fraction of droplet-
free $F$ in each vertical interval (e.g., if there are 90 cloud samples and 10 non-cloudy samples, $F = 10/(10+90) = 10\%$); TAS
and $f$ are the aircraft true air speed ($\sim 55$ m s$^{-1}$) and sampling frequency (10 Hz), respectively. The size of droplet-free air is
used as a proxy for the entrained air parcels' size. In equation (7), the time scale for a droplet of radius $r_{va}$ to completely
evaporate (evaporate time) is given by:
$$\tau_{evap} = -\frac{r_{va}^2}{2AS_0},$$  (10)
where $S_0$ is the supersaturation of the droplet-free air at the corresponding altitude (Yau and Rogers, 1996); $A$ is a affected by
air temperature and pressure (see Appendix B for details).

Another dynamical measure given by the ratio of $L^*$ to $\eta$ is transition scale number ($N_L$) (Lu et al. (2011)):
$$N_L = \frac{L^*}{\eta},$$  (11)





where transition length ($L^*$) is considered as the corresponding $L$ value when $Da = 1$ (Lehmann et al., 2009) and is given as
follows:
$$L^* = \varepsilon^{1/2} \tau_r^{3/2}.$$  (12)
In equation (11), $\eta$ is the Kolmogorov length scale (Wyngaard, 2010), which is given by:
$$\eta = (\frac{v^3}{\varepsilon})^{1/4},$$  (13)
where $v$ is the kinematic viscosity (Wyngaard, 2010). A higher probability of HM mixing corresponds to a larger value of $N_L$.
**3 Results**
**3.1 Entrainment-Mixing Mechanisms from the Microphysical and Dynamical Perspectives**
It has been known that it can be uncertain and even problematic to determine the representative adiabatic values from the
observational data needed in calculation of the above-mentioned microphysical measures (Yeom et al., 2017; Jensen et al.,
1985; Yum et al., 2015). For example, because vertical velocity and concentration of cloud condensation nuclei can change
spatially in clouds, $n_a$ and $r_{va}$ change accordingly. Entrainment-mixing in clouds adds difficulties to determine accurate values
of $r_{va}$, $n_a$ and $LWC_a$. Improper estimation of adiabatic values may violate the theoretical expectation: $n_a \geq n_h \geq n_c \geq n_i$ and $r_{va} \geq$
$r_v$, and then cause unrealistic HMDs. Different adiabatic variables have been used in previous studies. For example, the
maximum volume mean radius and number concentration are used as proxy values for $r_{va}$ and $n_a$ for each horizontal penetration,
respectively (Yeom et al., 2017; Yum et al., 2015); $LWC_a$ is calculated from the adiabatic growth from cloud base, and the
maximum number concentration of whole flight penetration is considered as $n_a$ (Burnet and Brenguier, 2007; Lehmann et al.,
2009); $n_a$ is the mean value of top 2% of $n_c$ for each flight and $r_{va}$ is calculated using adiabatic water vapor mixing ratio,
adiabatic total water mixing ratio and $n_a$ for a horizontal penetration (Small et al., 2013).

To examine the influence of using different adiabatic properties, we compare $\psi_i$ (i = 1, 4) calculated with different adiabatic
variables (Table 1) at each level near the stratiform cloud tops for the data collected during the four flights. Only the results
for the first microphysical measure are shown in Figure 3; the other results are shown in the Supporting Information. In Figure
3, $LWC_a$ is based on the adiabatic growth from cloud base, the maximum number concentration at each level is assumed as $n_a$,
and $r_{va}$ is calculated from $LWC_a$ and $n_a$. In Figure S1, $LWC_a$ is based on the adiabatic growth from cloud base, the maximum
volume mean radius at each level is assumed as $r_{va}$, and $n_a$ is calculated from $LWC_a$ and $r_{va}$. In Figure S2, the maximum liquid
water content at each level is assumed as $LWC_a$, the maximum number concentration at each level is assumed as $n_a$, and $r_{va}$ is
calculated from $LWC_a$ and $n_a$. In Figure S3, the maximum liquid water content at each level is assumed as $LWC_a$, the maximum


volume mean radius at each level is assumed as $r_{va}$, and $n_a$ is calculated from $LWC_a$ and $r_{va}$. In Figure S4, the maximum number
concentration at each level is assumed as $n_a$, the maximum volume mean radius at each level is assumed as $r_{va}$, and $LWC_a$ is
calculated from $n_a$ and $r_{va}$. According to the definitions, $\psi_i$ (i = 1, 4) are expected to range from 0 to 1. However, some values
of $\psi_i$ (i = 1, 4) are larger than 1 or smaller than 0 in Figure 3 and Figures S1 – S4, which could be caused by uncertainties in
$r_{va}$, $LWC_a$, $n_a$, and cloud base (Lu et al., 2014b; Lu et al., 2014a; Lu et al., 2013b). Furthermore, these figures suggest different
vertical distributions of HMDs for the same flight, suggesting that high sensitivity of the conventional HMDs to the methods
for determining the adiabatic values could pose a serious problem as to which figure represents the reality of entrainment-
mixing mechanisms.

Since the above analysis from the microphysical perspective does not tell a consistent story about the vertical variation of
HMD, $Da$ and $N_L$ are examined from the dynamical perspective. Figures 4 (a), (c), (e) and (g) show the height dependence of
$Da$ during each of the four flights. It is obvious that $Da$ decreases with decreasing altitudes. Figures 4 (b), (d), (f) and (h) show
a significant increasing trend of $N_L$ with decreasing altitudes. The method for setting the adiabatic values in Figure 4 is the
same as that in Figure 3, i.e., $LWC_a$ is based on the adiabatic growth from cloud base, the maximum number concentration at
each level is assumed as $n_a$, and $r_{va}$ is calculated from $LWC_a$ and $n_a$. Unlike the microphysical measures, vertical variation of
$Da$ or $N_L$ are similar when different methods for determining adiabatic values are used (Figures S5 – S8). It is expected that a
smaller $Da$ (larger $N_L$) represents a larger HMD. The results of $Da$ and $N_L$ both suggest more IM mixing closer to cloud top.
It is noteworthy that this result is robust, not affected by the methods for obtaining the adiabatic values, and thus should reflect
the real height dependence of entrainment-mixing mechanisms.

The different vertical distributions of HMDs and the inconsistency between microphysical HMDs and dynamical measures are
mainly due to the improper estimations of adiabatic values. For example, during the flight of 16 July in Figure 3, the HMDs
decrease with the decreasing altitudes, and most of the HMDs are negative. The negative values do not meet the theoretical
expectations and these trends are completely inconsistent with those of dynamical measures. The vertical variations of some
important properties of this case are shown in Figure 5. The negative values of HMDs are due to unexpected result of $r_{va} \le r_{vc}$.
Under these circumstances, the difference between $r_{vc}$ and $r_{va}$ becomes larger with the decreasing altitudes, corresponding to
the decreasing trends of HMDs with the decreasing altitudes. Besides the first method, the other four methods mentioned above
also have their own unreasonable points. For example, $r_{va} \le r_{vc}$ exists under the methods 1, 3 and 4; $n_a \le n_c$ exists under the
methods 2 and 4; $r_{va}$ does not always increase with the increasing altitudes under the methods 2, 4 and 5 (See figures S9 to
S13 for details). Overall, the inconsistency among the microphysical HMDs estimated with different methods to determine the
adiabatic variables calls for a new microphysical measure of entrainment-mixing mechanisms.





**3.2 New Microphysical Measure**
As discussed in Section 3.1, the existing microphysical measures of HMDs depend on the different adiabatic values to a great
extent. In order to avoid this kind of uncertainty, a new dimensionless HMD ($\psi_5$) is introduced to quantify the different
entrainment-mixing mechanisms:
$\psi_5 = dis(r_{vc}^3) / dis(\mathrm{LWC_c})$,                                                          (14)
where $dis$ represents the relative standard deviation expressed by the ratio of standard deviation to the average value over each
level. During entrainment-mixing and evaporation processes, $\mathrm{LWC_c}$ always decreases but $r_{vc}$ decreases in the HM mixing and
remains constant in the extreme IM mixing. Therefore, the extreme IM mixing corresponds to $\psi_5 = 0$, and the larger the value
of $\psi_5$ is, the more HM the entrainment mixing is. To make sure that $\psi_5$ is applied properly, the correlation between $r_{vc}^3$ and
$\mathrm{LWC_c}$ must be positive. If the correlation is negative, IM mixing with subsequent ascent is likely to occur (Lu et al., 2013a;
Lehmann et al., 2009; Wang et al., 2009; Siebert et al., 2006; Lasher‐trapp et al., 2005). It is worth mentioning that $\psi_5$ does
not require using adiabatic values, and thus can overcome the deficiencies of $\psi_i$ (i = 1, 4) associated with choosing different
adiabatic cloud properties.

The vertical variation of $\psi_5$ for the 4 flights are shown in Figure 6. The small value of $\psi_5$ near the cloud tops shows that
entrainment-mixing approaches extreme IM, consistent with conclusions in several previous studies based on the POST data
(Gerber et al., 2013; Gerber et al., 2016; Malinowski et al., 2013). The increase of $\psi_5$ with decreasing altitudes indicates that
the trends towards more HM with the decreasing altitudes, consistent with the results of $Da$ and $N_L$ (Figure 4 and Figures S5
– S8). We also check the relationship between $r_{vc}^3$ and $\mathrm{LWC_c}$ and the two quantities are positively correlated (not shown).

The relationships between $\psi_5$ versus $Da$ and $N_L$ of the 4 flights are shown in Figure 7 and are well fitted by the equations used
in Luo et al. (2020)
$\psi_5 = a_1 \exp(b_1 Da^{c_1})$,                                                                        (15)
$\psi_5 = a_2 \exp(b_2 N_L^{c_2})$,                                                                       (16)
where the parameters $a_1$ and $a_2$ are positive; $b_1$ and $b_2$ are negative; $c_1$ is positive and $c_2$ is negative. The negative correlation
of $\psi_5$ vs $Da$ and positive correlation of $\psi_5$ vs $N_L$ are evident and in keeping with theoretical arguments, suggesting that a smaller
$Da$ or a larger $N_L$ corresponds to a higher $\psi_5$. Such relationships further confirm the utility and applicability of $\psi_5$ in studying
entrainment-mixing mechanisms. The correlation coefficients of the linear regression of for $\psi_5$ vs $Da$ and $\psi_5$ vs $N_L$ are about
0.66 and 0.60, respectively, suggesting that $Da$ and $N_L$ are basically equivalent for understanding the entrainment-mixing
parameterization.






The equivalence of $Da$ and $N_{\mathrm{L}}$ is further supported by the tight negative correlation between $Da$ and $N_{\mathrm{L}}$ (Figure 8). Similar
results have been reported in Gao et al. (2018) using numerical simulations, and Desai et al. (2021) based on holographic
measurements. However, the underlying reasons are different. Figure 9 shows that $L$ and $L^*$ are negatively correlated, opposite
to the positive correlation between $L^*$ and the Taylor microscale in Gao et al. (2018); Taylor microscale is used as $L$ in the
calculation of $\tau_{\mathrm{mix}}$ in equation (8) in Gao et al. (2018). It is easy to derive from equations (7), (8), (10) and (11) that $Da : N_{\mathrm{L}} =$
$L : L^*$, others being equal:
$$\frac{Da}{N_{\mathrm{L}}} = \frac{-2AS_0\eta}{\varepsilon^{1/3}r_{\mathrm{va}}^2} \cdot \frac{L}{L^*} \qquad (17)$$
Therefore, as long as $L$ and $L^*$ are nearly linearly correlated, $Da$ and $N_{\mathrm{L}}$ are equivalent. When extreme IM mixing dominates
near cloud top, $\varepsilon$ is small (Figure 10), which mainly determines small $L^*$; $L$ is large near cloud top (Figure 10). Therefore, $L$
and $L^*$ are negatively correlated. The vertical distributions of affecting factors on entrainment-mixing are detailed in the next
sub-section.

**3.3 Further analysis of Affecting Factors**
According to the analyses in Sections 3.1 and 3.2, the dynamical and microphysical measures both indicate that entrainment-
mixing mechanisms change from IM to HM with decreasing altitudes. Here we provide the physical explanation for such
behavior under the framework of HM/IM entrainment-mixing mechanisms, by analyzing the vertical variations of all the
variables defining $Da$ and $N_{\mathrm{L}}$, i.e., $\varepsilon$, relative humidity (RH) and $L$.

First, Figures 10 (a), (d), (g) and (j) show that $\varepsilon$ increases with decreasing altitudes, which is opposite to that for cumulus
clouds (Small et al. (2013) and Jarecka et al. (2013)). According to definition of $Da$ (equation (7)) and $N_{\mathrm{L}}$ (equation (11)), the
increase of $\varepsilon$ leads to the decrease of $Da$ and increase of $N_{\mathrm{L}}$, others being equal. Therefore, $\varepsilon$ is an important factor to cause $Da$
to decrease and $N_{\mathrm{L}}$ to increase with the decreasing altitudes (Figure 4 and Figures S5 – S8).

Second, the vertical variation of entrainment-mixing can also be attributed to that of entrained air sizes. Figures 10 (b), (e), (h)
and (k) show that $L$ decreases significantly with decreasing altitudes, which leads to a decrease of $Da$ with decreasing altitudes
since $Da$ is proportional to $\tau_{\mathrm{mix}}$, and thus $L$. The importance of $L$ has rarely been studied in previous literatures for height
dependence of entrainment-mixing. The decrease of $L$ with decreasing altitudes agrees generally with the cascade of
breakdown of dry air parcels entrained at the cloud top.





Third, vertical variation of entrained air RH plays a significant part in determining the entrainment-mixing mechanisms. In
former literatures (Yeom et al., 2017; Lu et al., 2018), RH is commonly assumed to be constant across multiple different
altitudes when calculating $\tau_{evap}$ using $S_0$ = RH - 1. In fact, RH should not be a constant. We determine RH as the mean RH of
droplet-free air in each level. Figures 10 (c), (f), (i) and (l) show that RH increases with decreasing altitudes due to droplet
evaporation. According to the definition of $Da$, $Da$ decreases with the increase of $\tau_{evap}$, and thus decreases with the increase of
RH (equation (7) and (10)). Equations (10), (11) and (12) show that $N_L$ increases with increasing RH. Both $Da$ and $N_L$ indicate
more HM mixing at a lower altitude. These results suggest that the increases of $\varepsilon$ and RH and the decrease of $L$ with decreasing
altitudes are in keeping with the variation of entrainment-mixing processes, together playing the primary role in determining
the vertical distribution of HMD observed.

It is noted that, $r_{va}$ also affects $Da$ and $N_L$ through its effect on $\tau_{evap}$. However, $r_{va}$ depends on how adiabatic values are estimated
in Section 3.1 (Figure S9 – S14 in the Supporting Information). Therefore, the vertical variation of $r_{va}$ is not analyzed here. No
matter which method is used to determine the adiabatic values, the trends of vertical variation of $Da$ and $N_L$ do not change
(Section 3.1). The vertical variation of $Da$ and $N_L$ indicates the dominance of the combined effects of $\varepsilon$, RH and $L$ in
determining the vertical variation of entrainment-mixing processes from IM towards HM with decreasing altitudes.

These results are in keeping with the results drawn in Wang et al. (2009) and Yum et al. (2015) in the sense that a trait of IM
mixing is prevalent near cloud top but at mid-levels of clouds a trait of HM mixing becomes dominant, according to the
analysis of cloud microphysical relationships at different altitudes of marine stratiform clouds. However, there are big
differences in the spatial scale of analysis between our and their studies. We focus on near cloud top regions from cloud top to
where droplet-free air patches can still be found, mostly less than 100 m from cloud top (Figure 3). On the other hand, Yum et
al. (2015) and Wang et al. (2009) examined mid-levels of stratiform clouds where there remained no droplet-free air patches
as well as near cloud top regions. They suggested that the vertical variation of cloud microphysical properties relationships
could be caused by vertical circulation of diluted parcels affected by entrainment; the actual mixing near cloud top might have
been IM as $D_a$ and $N_L$ at this level suggested; as these parcels moved down, the droplets evaporated fast, resulting in cloud
microphysical relationships that would be explained as a trait of HM mixing.
**4 Concluding Remarks**
The observational data of marine stratiform clouds measured from aircraft during the campaign of Physics of Stratocumulus
Top (POST) are used to examine the height dependence of entrainment-mixing mechanisms. The sawtooth penetrations are
analyzed to acquire fine information on the vertical structure of entrainment-mixing near stratiform cloud tops, from the





microphysical and dynamical perspectives. To ensure high vertical resolution, we take 5 m as one altitude distance bin of all
sawtooth patterns for the four flights selected in this study.

From the microphysical perspective, the traditional homogeneous mixing degrees vary distinctly with the decreasing altitudes
due to different methods for obtaining adiabatic values. In order to overcome this difficulty, a new homogeneous mixing degree
describing the distributions of scatters in the mixing diagram is introduced to quantify different entrainment-mixing
mechanisms. The new homogeneous mixing degree is introduced by relative standard deviation of cubic volume mean radius
divided by relative standard deviation of liquid water content. If the new homogeneous mixing degree is larger, the mixing is
more likely to be homogeneous. The new measure increases with the decreasing altitudes, i.e., more homogeneous with
decreasing altitudes. This new measure is not affected by the methods for obtaining adiabatic values and shed new light on the
study of entrainment-mixing mechanisms.

From the dynamical perspective, Damkohler number decreases and transition scale number increases with decreasing altitudes.
The relationships between the new homogeneous mixing degree *vs*. Damkohler number and transition scale number are
negative and positive, respectively, consistent with theoretical expectation. Therefore, both microphysical and dynamical
analyses indicate the trends from inhomogeneous mixing to homogeneous mixing when altitude decreases.

The factors underlying the vertical variation of entrainment-mixing mechanisms are examined, including vertical distributions
of dissipation rate, size of droplet-free air and relative humidity in droplet-free air. Dissipation rate increases and droplet-free
air size decreases with the decreasing altitudes. Therefore, mixing is faster at the lower altitude and homogeneous mixing is
more likely to occur. Relative humidity increases with decreasing altitudes, which indicates that droplets are less likely to be
completely evaporated at the lower altitude. The combined effects of the three factors determine the entrainment-mixing
vertical evolution.

It is noteworthy that the traditional homogeneous mixing degrees are still useful properties to quantify entrainment-mixing
mechanisms, if adiabatic values of microphysical properties are properly determined. The new homogeneous mixing degree
defined here provides an alternative method to quantify entrainment-mixing mechanisms by overcoming difficulties of
determining adiabatic microphysical properties needed in the traditional approaches. This new method can be applied to other
datasets since the new definition is based on theoretical understanding of entrainment-mixing mechanisms, which is not limited
to the dataset used here. It would be interesting to apply this method to other stratocumulus and cumulus observations in
different climate zones.



**Code and Data Availability**

The codes can be accessed by contacting Chunsong Lu via luchunsong110@gmail.com. The POST data is available on https://archive.eol.ucar.edu/projects/post/.

**Author Contributions**

SG performed the data analysis and manuscript writing. CL proposed the idea, guided this work and modified the manuscript. YL and SSY supervised this work and helped revise the manuscript. JZ and LZ offered helps to the data analysis. ND and YM also contributed to the modification of manuscript.

**Competing Interests**

The authors declare that they have no conflict of interest.





**Appendix A**
Turbulent dissipation rate ($\varepsilon$) is calculated by three dimensional wind velocities (Meischner et al., 2001)
$$\varepsilon \approx \frac{D_{NN}^{3/2}}{(4.01m)^{3/2}d},$$

366        (A1)

with $m \approx 0.2(2\pi)^{2/3}$ (Panofsky, 1984). $D_{NN}$ is the local spatial structure function using three wind components and is defined
as:
$$D_{NN}(t,d) = \frac{1}{3}\{\frac{8}{7}[u(t)-u(t-\frac{d}{TAS})]^2 + \frac{8}{7}[v(t)-v(t-\frac{d}{TAS})]^2 + [w(t)-w(t-\frac{d}{TAS})]^2\},$$    (A2)
where three wind components, east, north and vertical, are represented by $u$, $v$ and $w$, respectively; TAS is the aircraft true air
speed (~55m s$^{-1}$); $t$ is the time; $d$ is the scale parameter:
$$d = TAS\Delta t .$$    (A3)
where $\Delta t$ is the time interval, which is set to 0.1 s.





**Appendix B**
The parameter $A$ in equation (10) is
$$A = \frac{1}{[(\frac{L_h}{R_v T} - 1)\frac{L_h \rho_L}{KT} + \frac{\rho_L R_v T}{D e_s(T)}]}, \tag{B1}$$

where $R_v$, $L_h$, $T$, $K$, $\rho_L$, $D$, and $e_s(T)$ are water vapor specific gas constant, latent heat, temperature, coefficient of air thermal
conductivity coefficient, liquid water density, water vapor diffusion coefficient in air and vapor pressure of saturation,
respectively.



**Acknowledgment**
The authors thank the crew of the POST campaign. This research was supported by the National Natural Science Foundation
of China (41822504, 42027804, 42075073, 41975181), the National Key Research and Development Program of China
(2019YFA0606803) and Qinglan Project (R2018Q05).



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



**Table 1.** List of different methods determining adiabatic values

| Number | Methods |
|---|---|
| 1 | LWC$_a$: calculated from the adiabatic growth from cloud base;<br><br>$n_a$: maximum number concentration in each level;<br><br>$r_{va}$: calculated by $r_{va} = \sqrt[3]{\dfrac{LWC_a}{\frac{4}{3}\pi\rho_L n_a}}$ . |
| 2 | LWC$_a$: calculated from the adiabatic growth from cloud base;<br><br>$r_{va}$: maximum volume mean radius in each level;<br><br>$n_a$: calculated by $n_a = \dfrac{LWC_a}{\frac{4}{3}\pi\rho r_{va}^3}$ . |
| 3 | LWC$_a$: maximum liquid water content in each level<br><br>$n_a$: maximum number concentration in each level;<br><br>$r_{va}$: calculated by $r_{va} = \sqrt[3]{\dfrac{LWC_a}{\frac{4}{3}\pi\rho n_a}}$ . |
| 4 | LWC$_a$: maximum liquid water content in each level;<br><br>$r_{va}$: maximum volume mean radius in each level;<br><br>$n_a$: calculated by $n_a = \dfrac{LWC_a}{\frac{4}{3}\pi\rho r_{va}^3}$ . |
| 5 | $n_a$: maximum number concentration in the interval;<br><br>$r_{va}$: maximum volume mean radius in the interval;<br><br>LWC$_a$: calculated by $LWC_a = \dfrac{4}{3}\pi\rho r_{va}^3 n_a$ . |






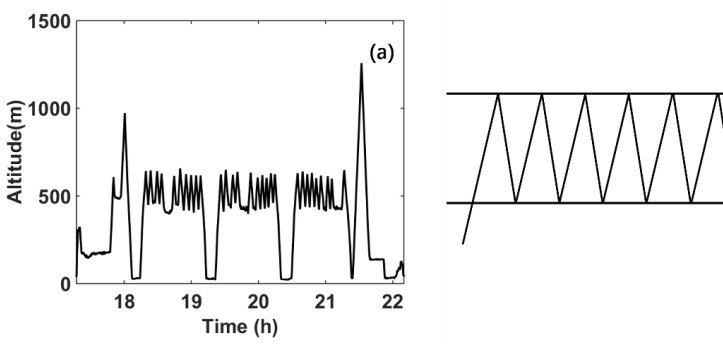


**Figure 1.** (a) Flight track on 16 July 2008. (b) Altitude stratification procedure of the sawtooth patterns, with the mean vertical resolution of

5 m such that $\Delta h_1 = \Delta h_2 = \cdots = \Delta h_n = 5$ m.

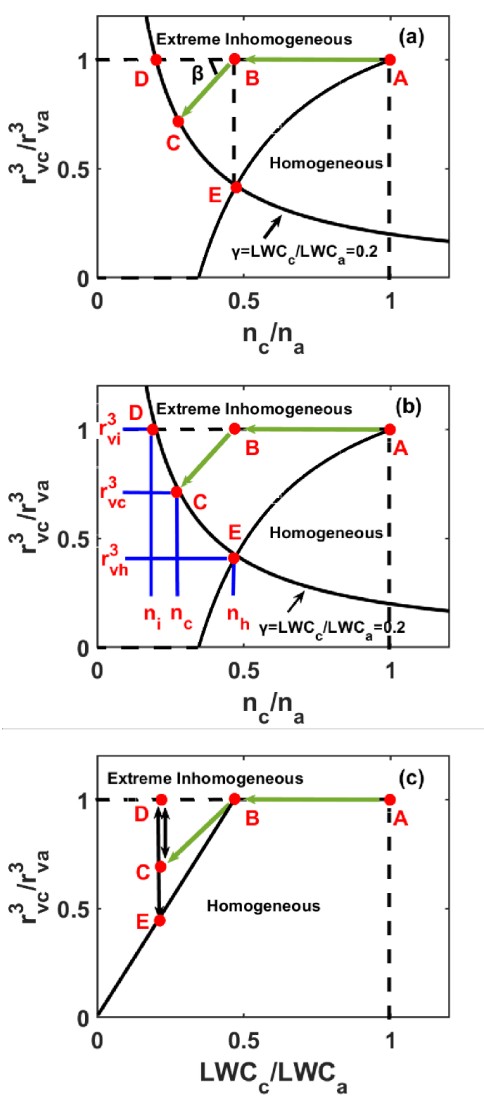


**Figure 2.** Microphysical diagram interpretating the definition for different homogeneous mixing degrees ((a) $\psi_1$; (b) $\psi_2$, $\psi_3$; (c) $\psi_4$). The

Points A and B represent the adiabatic state and the state after entrainment, respectively. If the extreme inhomogeneous mixing process

occurs, the cloud state approaches Point D; if the homogeneous mixing process occurs, the cloud state approaches Point E. The actual mixing

and evaporation processes are between the two extremes and cloud state approaches Point C. Extreme inhomogeneous mixing process is

represented by the horizontal dashed line; homogeneous mixing process is represented by the solid line starting from Point A in (a) and (b),

and the solid line starting from Point B in (c). Another black line in (a) and (b) corresponds to contour of $\gamma = 0.2$ defined as the ratio of liquid

water content (LWC$_c$) to its adiabatic value (LWC$_a$). The vertical dashed line represents the x-axis property equal to 1. The horizontal blue

solid lines represent the y-axis properties of Point D ($r_{vi}^3$), Point C ($r_{vc}^3$) and Point E ($r_{vh}^3$). The vertical blue solid lines represent the x-axis

properties of Point D ($n_i$), Point C ($n_c$) and Point E ($n_h$). See text for the meanings of other symbols.

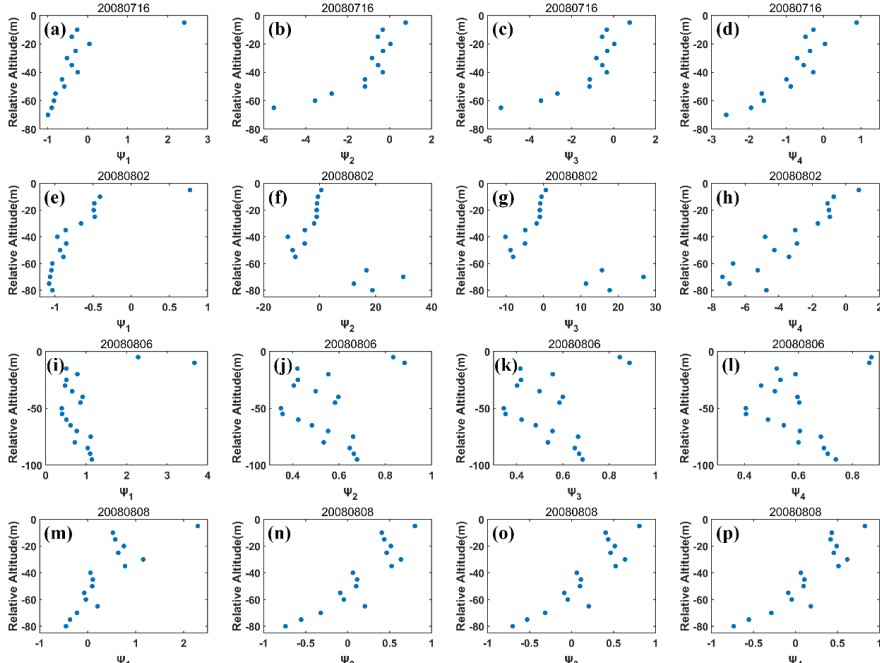

548

**Figure 3.** Height dependence of the first homogeneous mixing degree ($\psi_1$) on (a) 16 July 2008, (e) 02 August 2008, (i) 06 August 2008 and

(m) 08 August 2008; height dependence of the second homogeneous mixing degree ($\psi_2$) on (b) 16 July 2008, (f) 02 August 2008, (j) 06

August 2008 and (n) 08 August 2008; height dependence of the third homogeneous mixing degree ($\psi_3$) on (c) 16 July 2008, (g) 02 August

2008, (k) 06 August 2008 and (o) 08 August 2008; and the fourth homogeneous mixing degree ($\psi_4$) on (d) 16 July 2008, (h) 02 August 2008,

(l) 06 August 2008 and (p) 08 August 2008. The relative altitude on the y-axis equal to 0 represents the cloud tops. Adiabatic liquid water

content ($LWC_a$) is obtained by the adiabatic growth from cloud base, adiabatic number concentration ($n_a$) is assumed to be the maximum

volume mean radius at each level, and adiabatic volume mean radius ($r_{va}$) is calculated with $LWC_a$ and $r_{va}$.



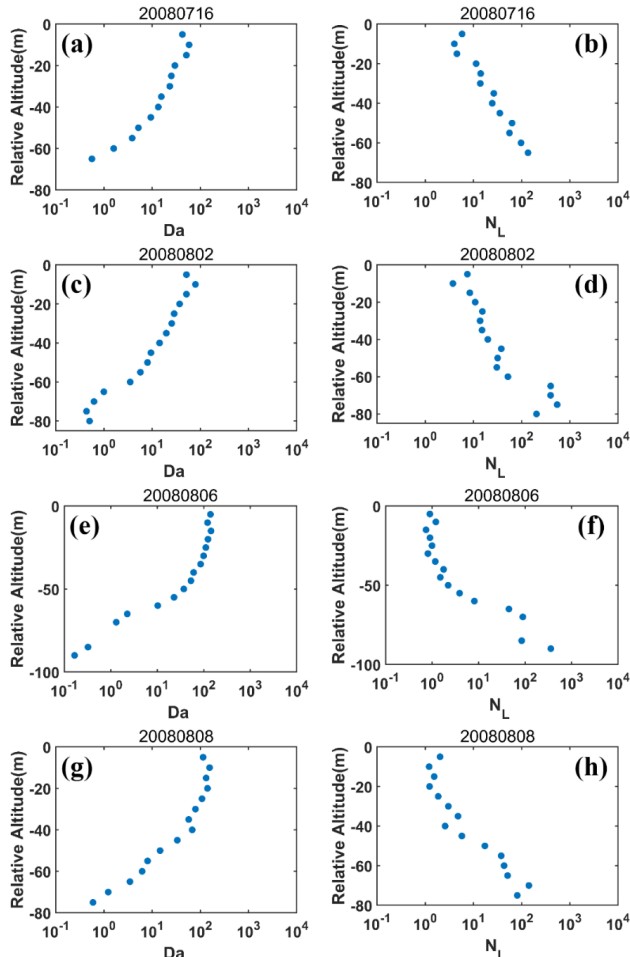

556

**Figure 4.** Height dependence of Damkohler number (*Da*) on (a) 16 July 2008, (c) 02 August 2008, (e) 06 August 2008 and (g) 08 August

2008; height dependence of transition scale number ($N_L$) on (b) 16 July 2008, (d) 02 August 2008, (f) 06 August 2008 and (h) 08 August

2008. The relative altitude on the y-axis equal to 0 represents the cloud tops. Adiabatic liquid water content (LWC$_a$) is obtained by the

adiabatic growth from cloud base, adiabatic number concentration ($n_a$) is assumed to be the maximum volume mean radius at each level,

and adiabatic volume mean radius ($r_{va}$) is calculated with LWC$_a$ and $r_{va}$.



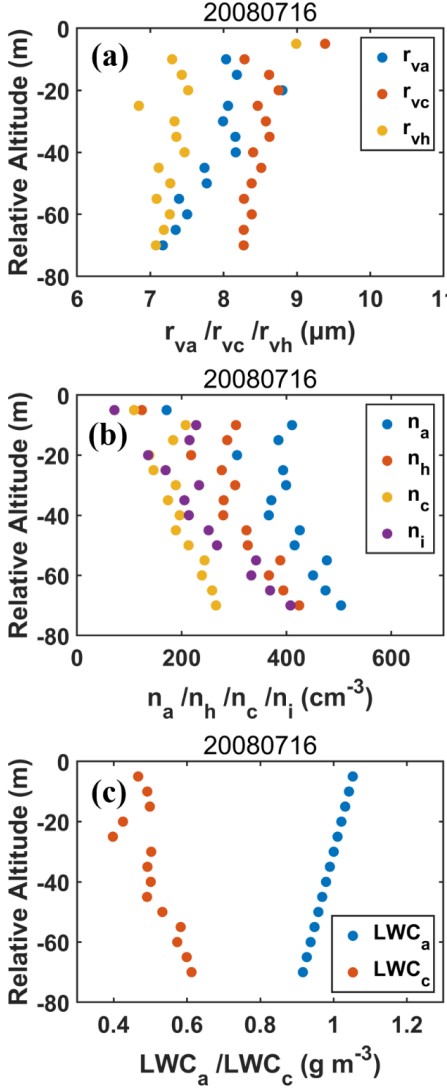

562

**Figure 5.** Height dependence of (a) $r_{va}$, $r_{vc}$, $r_{vh}$, (b) $n_a$, $n_h$, $n_c$, $n_i$ and (c) LWC$_a$, LWC$_c$ on 16 July 2008. The relative altitude on the y-axis

equal to 0 represents the cloud tops. Adiabatic liquid water content (LWC$_a$) is obtained by the adiabatic growth from cloud base, the maximum

number concentration at each level is assumed to be adiabatic number concentration ($n_a$), and adiabatic volume radius ($r_{va}$) is calculated with

LWC$_a$ and $n_a$.





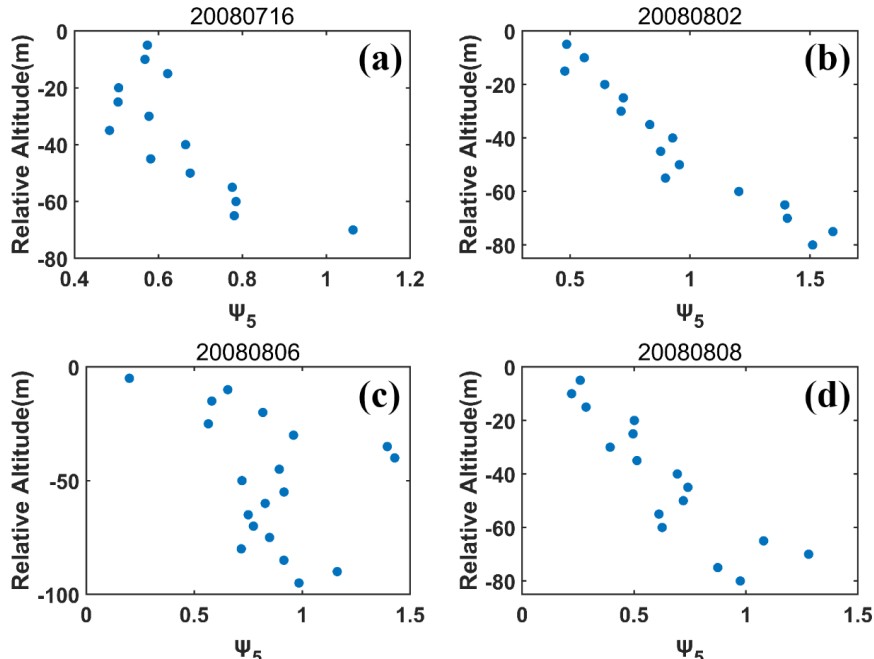

**Figure 6.** Height dependence of the newly defined homogeneous mixing degree ($\psi_5$) on (a) 16 July 2008, (b) 02 August 2008, (c) 06 August

2008 and (d) 08 August 2008. The relative altitude on the y-axis equal to 0 represents the cloud tops.




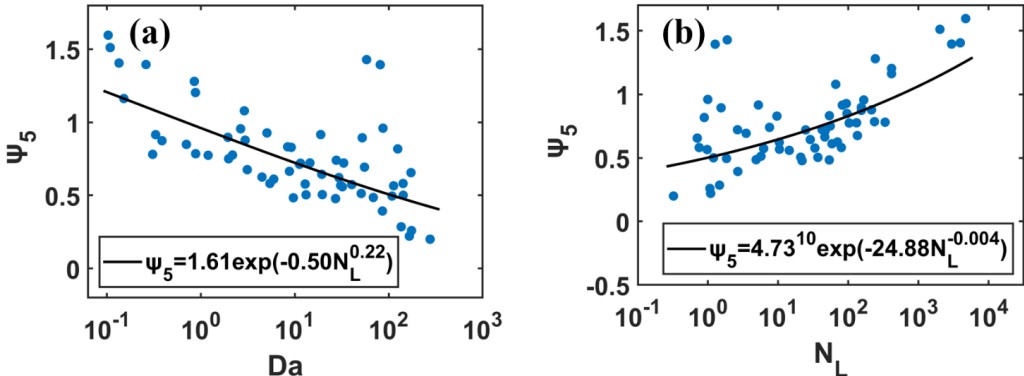


**Figure 7.** Relationships of the newly defined homogeneous mixing degree ($\psi_5$) versus (a) Damkohler number (*Da*) and (b) transition scale

number ($N_L$).





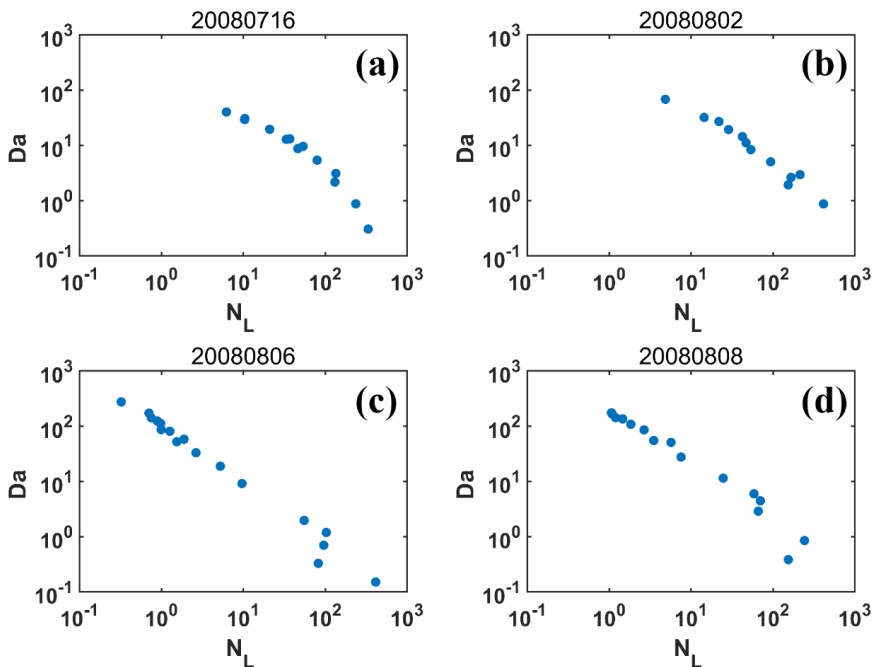


**Figure 8.** Relationships of Damkohler number (*Da*) versus transition scale number ($N_L$) on (a) 16 July 2008, (b) 02 August 2008, (c) 06

August 2008 and (d) 08 August 2008.





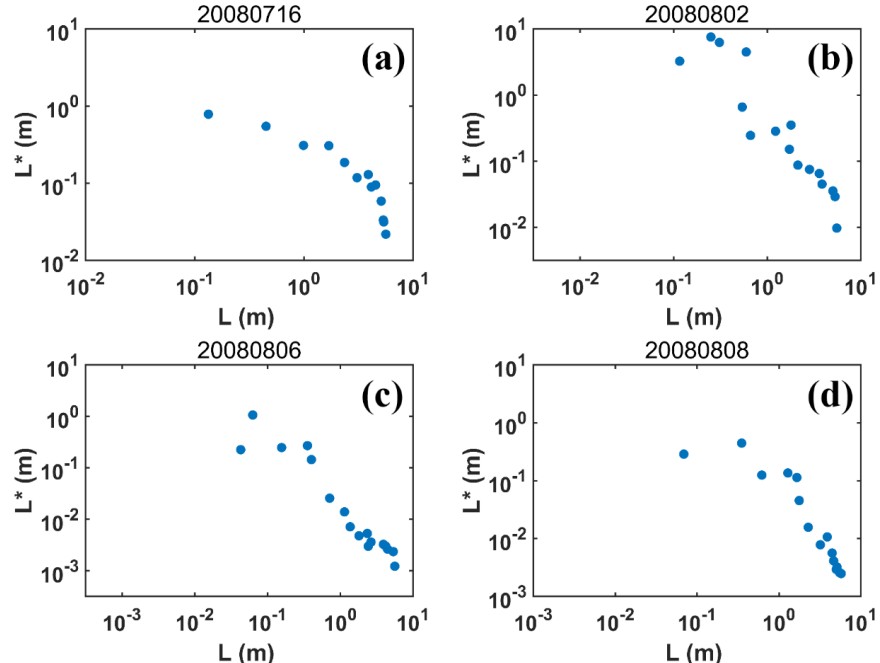

**Figure 9.** Relationships of transitional scale ($L^*$) versus droplet-free air length ($L$) on (a) 16 July 2008, (b) 02 August 2008, (c) 06 August 2008 and (d) 08 August 2008.

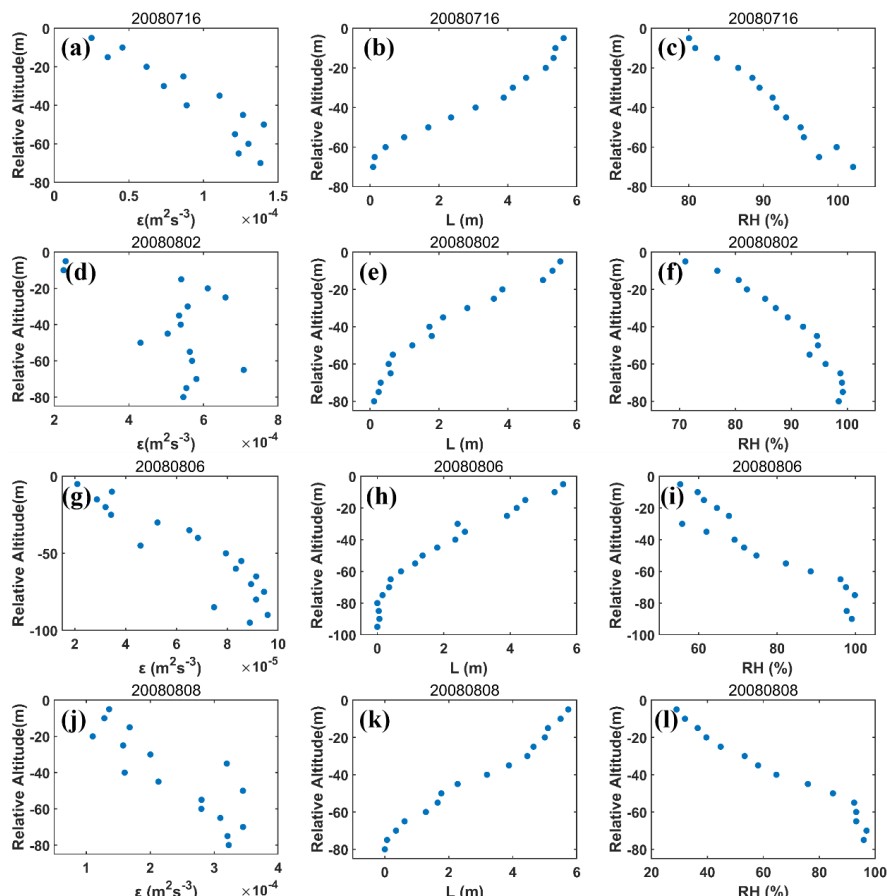

**Figure 10.** Height dependence of dissipation rate ($\varepsilon$) on (a) 16 July 2008, (d) 02 August 2008, (g) 06 August 2008 and (j) 08 August 2008; height dependence of relative humidity (RH) of droplet-free air on (b) 16 July 2008, (e) 02 August 2008, (h) 06 August 2008 and (k) 08 August 2008; and height dependence of length of droplet-free air ($L$) on (c) 16 July 2008, (f) 02 August 2008, (i) 06 August 2008 and (l) 08 August 2008. The relative altitude on the y-axis equal to 0 represents the cloud tops.