# Peer review of "Comprehensive Quantification of Height Dependence of"

_Atmospheric Chemistry and Physics, 2021_

## Referee Comment (RC2)

**Review Report**

**Title**: Comprehensive  Quantification of Height Dependence of Entrainment-Mixing between Stratiform Cloud Top and Environment

**Author**:
Sinan Gao, Chunsong Lu, Yangang Liu, Seong Soo Yum, Jiashan Zhu, Lei Zhu, Neel Desai, Yongfeng Ma

**Summary**:

This study propose a new measure for the homogeneous mixing degree which is independent of any adiabatic values. This measure was developed using observational data of marine stratiform clouds measured from aircraft during the campaign of Physics of Stratocumulus Top (POST) data. It is proposed that new method of mixing degree can be alternative method to quantify entrainment-mixing mechanisms by overcoming difficulties of determining adiabatic microphysical properties needed in the traditional approaches.

**Comments:**

The paper is written well and provide another way to quantify the degree of mixing.  It is worth to publish in ACP with minor corrections given below.

1. **Regarding observation:**
   (i)  Cloud droplet size distribution is measured by CAS probe however, the particle size range is CAS is not mentioned such 0.5$\rightarrow$ 50 µm.
   (ii) CAS has disadvantage of large size bin width (about 10 µm) for above 20 µm particle diameter, therefore, does not give accurate size resolution. On the other hand, CDP or FSSP probe has better size resolution. Why did the authors choose CAS probe for this study?.

2.  **New microphysical measure of mixing degree:**
   The formula (14), at line 233,  for new microphysical measure should be discussed in data and method section along with other methods.
   This new method does not give any theoretical basis like the other mixing degree methods. This is a relative measure of HMD as deviation from the extremely inhomogeneous mixing line. But, does not quantify the amount of homogeneous mixing precisely.  A critical value for homogeneous mixing cannot be inferred from this method. This is a disadvantage of this method.
   Furthermore, the standard deviation of mean radius and LWC increases due to differences in mixing states (having different history of mixing) and in-cloud activation of CCN. These points should be discussed properly in the results.

3. Although, there have been several reports on HMD using in situ observations, a limitation of such quantification is missing. Like Khain et al. 2018 pointed out the drawback of mixing diagram to quantify HMD using in situ observations due to transient mixing state. Some discussion is needed on this point.

**Other minor corrections are**

4. Line 88: Sentence is not clear.
5. Line 135: eq (5):  Express the log values clearly for example ln (nc)

---

## Author Comment (AC1)

**Reply to Referee #1**

This is a nice study that comprehensively investigates the dependence of entrainment-mixing processes on the height in stratocumulus clouds using high-resolution aircraft data. There are two important contributions: (1) It develops a new method for determining the microphysical homogeneous mixing degree that addresses the conventional challenge of determining adiabatic properties, and (2) It combines various measures of microphysical homogeneous mixing degree to conclusively demonstrate the height dependence and explains the physical reasons. The manuscript is well written and the methodology is sound. I only have one concern about the influence of turbulent dissipation rate on the findings: its contribution should be strongly dependent on the decoupling state of the boundary layer, which has not been considered in this study (see details below). After relevant discussions (or analyses) are added, this manuscript should be accepted for publication in ACP.

Major comments:

It is reasonable that the turbulent dissipation decreases with altitude in the cloud layer, which has been proven in prior LES studies. However, the monotonic decrease (from cloud base to top) only occurs for the well-mixed stratocumulus-topped boundary layer. In decoupled conditions, the dissipation rate tends to decrease near the cloud base so that it should maximize somewhere in the middle of the cloud layer. There are two sources of evidence supporting this argument: one from LES (Stevens 2000) and one from observations (Zheng et al., 2016).

Stevens (2000) shows that the boundary layer decoupling causes a decrease of TKE (turbulent kinetic energy) near the cloud base, leading to a local minimum near the cloud base and a maximum in the middle of the cloud layer. (In the TKE equation, turbulent dissipation roughly balances the kinetic energy so the profiles of TKE can be used to infer the profile of turbulent dissipation). This is also demonstrated in the observations by Zheng et al. (2016) who found a significant role of decoupling in weakening the cloud-base updrafts.

Therefore, it is reasonable to conjecture that the contribution of the turbulent dissipation to the findings (height dependence of mixing) may vary in decoupled boundary layers. Since the clouds in this study were sampled near the coast of CA, they must be coupled, which explains the monotonic decrease of dissipation with height. But decoupled clouds should be very common in the downstream regions (e.g., Bretherton and Wyant, 1997) and midlatitudes (e.g., Zheng et al., 2020). Therefore, it worths more discussion on the probable modification of the results in other regions.

References:

Stevens, B., 2000. Cloud transitions and decoupling in shear-free stratocumulus-topped boundary layers. Geophysical research letters, 27(16), pp.2557-2560.

Zheng, Y., Rosenfeld, D. and Li, Z., 2016. Quantifying cloud base updraft speeds of marine stratocumulus from cloud top radiative cooling. Geophysical Research Letters, 43(21), pp.11-407.

Bretherton, C.S. and Wyant, M.C., 1997. Moisture transport, lower-tropospheric stability, and decoupling of cloud-topped boundary layers. Journal of Atmospheric Sciences, 54(1), pp.148-167.

Zheng, Y., Rosenfeld, D. and Li, Z., 2020. A more general paradigm for understanding the decoupling of stratocumulus-topped boundary layers: The importance of horizontal temperature advection. Geophysical Research Letters, 47(14), p.e2020GL087697.

**Reply:** Thank you very much for your recognition of our work. Yes, we agree that the contribution of dissipation rate should be strongly dependent on the decoupling state of the boundary layer, and we have added the relevant discussion in the manuscript Lines 280-288: "The clouds were sampled in the vicinity of the coast of Santa Cruz/Monterey, California, therefore, these clouds were well-mixed and coupled, which explains the monotonic decrease of $\varepsilon$ with the increasing height (Jones et al., 2011;Shupe et al., 2013). Note that the decoupled clouds should be very common in the downstream regions (e.g., Bretherton and Wyant, 1997) and midlatitudes (e.g., Zheng et al., 2020). The boundary layer decoupling causes a decrease of turbulent kinetic energy near the cloud base, leading to a local minimum near the cloud base and a maximum in the middle of cloud layer. The profile of turbulent kinetic energy can be used to infer the profile of $\varepsilon$ (Stevens, 2000). This is also demonstrated in the observations by Zheng et al. (2016) who found a significant role of decoupling in weakening the cloud-base updrafts. Therefore, in the future studies of decoupled stratocumulus in other regions, the results about entrainment-mixing mechanisms could be different due to the non-monotonic vertical variation of $\varepsilon$."

---

## Author Comment (AC2)

**Response to Referee #2**

**Title:**

Comprehensive Quantification of Height Dependence of Entrainment-Mixing between Stratiform Cloud Top and Environment

**Author:**

Sinan Gao, Chunsong Lu, Yangang Liu, Seong Soo Yum, Jiashan Zhu, Lei Zhu, Neel Desai, Yongfeng Ma

**Summary:**

This study proposes a new measure for the homogeneous mixing degree which is independent of any adiabatic values. This measure was developed using observational data of marine stratiform clouds measured from aircraft during the campaign of Physics of Stratocumulus Top (POST) data. It is proposed that new method of mixing degree can be alternative method to quantify entrainment-mixing mechanisms by overcoming difficulties of determining adiabatic microphysical properties needed in the traditional approaches.

**Comments:**

The paper is written well and provide another way to quantify the degree of mixing. It is worth to publish in ACP with minor corrections given below.

> **Reply**: Thank you very much for appreciating the importance of our work!

1. Regarding observation:
   - (i)    Cloud droplet size distribution is measured by CAS probe however, the particle size range in CAS is not mentioned such 0.5-50 μm.
   - (ii)   CAS has disadvantage of large size bin width (about 10 μm) for above 20 μm particle diameter, therefore, does not give accurate size resolution. On the other hand, CDP or FSSP probe has better size resolution. Why did the authors choose CAS probe for this study?

   **Reply**:
   - (i)    We have added the particle size range in CAS in Lines 88-90: "The Cloud and Aerosol Spectrometer (CAS) probe measured size distributions in the radius range of 0.29 - 25.5 μm at the frequency of 10 Hz. The data in the radius range of 1.0 - 25.5 μm are used to calculate microphysical properties, i.e., number concentration ($n_c$), liquid water content ($LWC_c$) and volume mean radius ($r_{vc}$)."
   - (ii)   Yes, we agree with the reviewer that the CAS probe has disadvantage of large size bin width (about 10 μm) for above 20 μm particle diameter. However, only the data from the CAS probe are available in this field campaign. Therefore, we use the CAS probe for this study. In future

studies, we will analyze different datasets with CDP and FSSP. We add some discussion (Lines 364 - 366): "This new method can be applied to other datasets with different cloud droplet size probes (e.g., the Forward Scattering Spectrometer Probe, FSSP), since the new definition is based on theoretical understanding of entrainment-mixing mechanisms, which is not limited to the dataset used here."

2. New microphysical measure of mixing degree:
The formula (14), at line 233, for new microphysical measure should be discussed in data and method section along with other methods.
**Reply**: According to the reviewer's comment, the formula for the new microphysical measure is discussed in data and method section (Lines 142 - 147): "A new dimensionless HMD ($\psi_5$) is introduced to quantify the different entrainment-mixing mechanisms:

$$\psi_5 = dis(r_{vc}{}^3)/dis(\mathrm{LWC}_c),\tag{7}$$

where *dis* represents the relative standard deviation expressed by the ratio of standard deviation to the average value over each level. During entrainment-mixing and evaporation processes, $\mathrm{LWC}_c$ always decreases but $r_{vc}$ decreases in the HM mixing and remains constant in the extreme IM mixing. Therefore, the extreme IM mixing corresponds to $\psi_5 = 0$, and the larger the value of $\psi_5$ is, the more HM the entrainment mixing is. More discussions on $\psi_5$ are given in Section 3.2."

This new method does not give any theoretical basis like the other mixing degree methods. This is a relative measure of HMD as deviation from the extremely inhomogeneous mixing line. But, does not quantify the amount of homogeneous mixing precisely. A critical value for homogeneous mixing cannot be inferred from this method. This is a disadvantage of this method.
**Reply**: We agree with the reviewer that the newly defined measure is a relative measure of homogeneous mixing degree (HMD) as deviation from the extremely inhomogeneous mixing line, but does not quantify the amount of homogeneous mixing precisely. This is indeed a disadvantage of this method. We have added the above discussions in the revised manuscript (Lines 356 - 358): "The new homogeneous mixing degree defined here is a relative measure of homogeneous mixing degree as deviation from the extremely inhomogeneous mixing line, but does not quantify the amount of homogeneous mixing precisely."

Furthermore, the standard deviation of mean radius and LWC increases due to differences in mixing states (having different history of mixing) and in-cloud activation of CCN. These points should be discussed properly in the results.
**Reply**: Yes, we have added the related discussion in Lines 358 - 361: "The relative dispersion of volume-mean radius and liquid water content increases due to differences in mixing states (Khain et al., 2018) and in-cloud activation of cloud

condensation nuclei (Derksen et al., 2009), which affects the calculation of the new homogeneous mixing degree."

3. Although, there have been several reports on HMD using in situ observations, a limitation of such quantification is missing. Like Khain et al. 2018 pointed out the drawback of mixing diagram to quantify HMD using in situ observations due to transient mixing state. Some discussion is needed on this point.
   **Reply**: Thank you very much for pointing out this. According to the comment, we have added some discussion (Lines 361 - 364): "As pointed out by Khain et al. (2018), the mixing diagram has limitations when it is applied to analyze entrainment-mixing mechanisms using in situ observations, due to transient mixing states."

Other minor corrections are
4. Line 88: Sentence is not clear.
   **Reply**: The sentence is revised (Line 88 - 90): "The Cloud and Aerosol Spectrometer (CAS) probe measured size distributions in the radius range of 0.29 - 25.5 μm at the frequency of 10 Hz. The data in the radius range of 1.0 - 25.5 μm are used to calculate microphysical properties, i.e., number concentration ($n_c$), liquid water content ($LWC_c$) and volume mean radius ($r_{vc}$)."

5. Line 135: eq (5): Express the log values clearly for example ln (nc)
   **Reply**: The log values are clearly expressed in Eq (5), according to the comment:

$$\psi_3 = \frac{\ln(n_c) - \ln(n_i)}{\ln(n_h) - \ln(n_i)} = \frac{\ln(r_{vc}^3) - \ln(r_{va}^3)}{\ln(r_{vh}^3) - \ln(r_{va}^3)} . \tag{R1}$$